# Recent Advances in ^64^Cu/^67^Cu-Based Radiopharmaceuticals

**DOI:** 10.3390/ijms24119154

**Published:** 2023-05-23

**Authors:** Olga O. Krasnovskaya, Daniil Abramchuck, Alexander Erofeev, Peter Gorelkin, Alexander Kuznetsov, Andrey Shemukhin, Elena K. Beloglazkina

**Affiliations:** 1Chemistry Department, Lomonosov Moscow State University, Leninskie Gory, 1/3, 119991 Moscow, Russia; abr_daniil@mail.ru (D.A.);; 2Research Laboratory of Biophysics, National University of Science and Technology (MISIS), Leninskiy Prospect 4, 119049 Moscow, Russia; 3Skobeltsyn Institute of Nuclear Physics, Lomonosov Moscow State University, Leninskie Gory, 1, Bld. 2, 119991 Moscow, Russia; 4Department of Physics, Lomonosov Moscow State University, Leninskie Gory, 1/2, 119991 Moscow, Russia

**Keywords:** copper-64, copper-67, PET, SPECT, radiotherapy, radioimmunotherapy

## Abstract

Copper-64 (T_1/2_ = 12.7 h) is a positron and beta-emitting isotope, with decay characteristics suitable for both positron emission tomography (PET) imaging and radiotherapy of cancer. Copper-67 (T_1/2_ = 61.8 h) is a beta and gamma emitter, appropriate for radiotherapy β-energy and with a half-life suitable for single-photon emission computed tomography (SPECT) imaging. The chemical identities of ^64^Cu and ^67^Cu isotopes allow for convenient use of the same chelating molecules for sequential PET imaging and radiotherapy. A recent breakthrough in ^67^Cu production opened previously unavailable opportunities for a reliable source of ^67^Cu with high specific activity and purity. These new opportunities have reignited interest in the use of copper-containing radiopharmaceuticals for the therapy, diagnosis, and theranostics of various diseases. Herein, we summarize recent (2018–2023) advances in the use of copper-based radiopharmaceuticals for PET, SPECT imaging, radiotherapy, and radioimmunotherapy.

## 1. Introduction

Nuclear medicine is based on the use of radiochemical properties of isotopes for the therapy and diagnosis of various diseases. Radiotherapy accompanies almost 50% of the use of chemotherapy, thus being an extremely important treatment modality not only in the treatment of tumor diseases but also in the palliative care of inoperable patients [1]. Since radiation therapy affects cancer tissues via DNA damaging, targeted, and strictly localized effects of radiation, preliminary imaging of the biodistribution of radiopharmaceuticals using theranostic pairs is extremely important [2].

A classical targeted radiopharmaceutical is a tissue-affine molecule, conjugated with chelator, which is further radiolabeled with radioactive isotope. Depending on the type of radiation emitted, imaging or therapeutic capability is assigned to the radiopharmaceutical category. Positron-emitting isotopes are used for PET imaging, and gamma-emitting radioisotopes are used for SPECT visualization; for radiotherapy, α-, β-, and Auger electron emitters are considered [3,4]. However, the design of radiopharmaceuticals is based not only on the emission properties of the isotope but also on the method of its synthesis, and the ability to produce the isotope in sufficient quantity, purity, and specific activity is extremely important. Radionuclides can be produced via nuclear reactors, linear accelerators, and medical cyclotrons, and can also be conveniently eluted from portable nuclide generators. However, the production of nuclides via nuclear reactors is a rather difficult task due to the large amount of radioactive waste produced along the way; moreover, the transportation of nuclear waste is a public safety issue. The use of cyclotrons allows the production of high-quality nuclides; however, a limited number of cyclotrons poses logistical problems for the delivery of short-lived radionuclides. The use of linear accelerators does not always make it possible to obtain a radionuclide of sufficient purity and activity [5,6,7]. We have summarized widely used radioisotopes for imaging and radiotherapy, generation methods, as well as drugs based on them in Table 1.

Radioligand theranostic treatment is based on the use of “theranostic pairs”, namely, on two radiopharmaceuticals with the same molecular target, which are differentially labeled with imaging/therapy radionuclides, thus allowing for sequential PET/SPECT-imaging and radiotherapy [59]. A successive administration of imaging (β^+^ or γ^−^ emitters) and therapeutic (β^−^, α^−^ or low-energy Auger/conversion electron emitters) radioisotopes make it possible to measure the drug uptake via PET/SPECT imaging and the following dosimetric calculation. The use of radioisotopes of the same element for both pre-imaging and therapy allows for an accurate radiation dose, while the use of diagnostic and therapeutic nuclides of different elements complicates both dosimetry and the pharmacokinetic study of a therapeutic radiopharmaceutical.

Several theranostic pairs, such as ^68^Ga/^177^Lu, ^124^I/^131^I, ^64^Cu/^67^Cu, ^43^Sc/^44^Sc/^47^Sc, ^83^Sr/^89^Sr, ^86^Y/^90^Y, ^110^In/^111^In, ^152^Tb/^161^Tb and ^90^Y/^177^Lu have been reported [60]. However, these theranostic couples have several disadvantages. The use of the ^68^Ga/^177^Lu theranostic pair is limited by the fact that gallium-68 generators are gradually being phased out by more cost-effective accelerator-based production, due to higher amounts of specific activity and no waiting required between productions, unlike the required 3–4 h interval between isotope elutions from portative generators [61]. ^124^I, widely used in medical practice, has an accompanying positron gamma radiation, which complicates PET diagnosis due to false signal emerging and distortion of PET results [62,63]. In addition, the fact that iodine is an easily removable leaving group leads to dehalogenation reactions within the body (for example, in the diagnosis of prostate cancer [64]). 

Copper is a metal essential to human health, and it is a constituent of many enzymes [65]. Natural copper has two stable isotopes, ^63^Cu and ^65^Cu, and five radioisotopes, ^60^Cu, ^61^Cu, ^62^Cu, ^64^Cu and ^67^Cu. Two of these isotopes, namely ^64^Cu and ^67^Cu, are considered therapeutic β-emitters for targeted radiation therapy. However, among all copper radioisotopes, ^64^Cu is the “golden mean” for PET imaging due to its unique emission properties.

Copper-64 (T_1/2_ = 12.7 h; β^+^: 18%, β^−^: 39%) possesses an attractive decay profile for nuclear medicine, which provides the possibility of using ^64^Cu-based radiopharmaceuticals for both PET imaging and radionuclide therapy. The combination of β^+^ and β^−^ emission imparts a high local radiation dose at the cellular level, and electron capture decay is accompanied by the Auger electrons’ emission, which also contributes to cytotoxic potency. 

^64^Cu is most often produced via the ^64^Ni(p,n) ^64^Cu reaction on biomedical cyclotrons. Typically, with a bombardment time of 4 hours of 40 mg ^64^Ni target yields 18.5 GBq of ^64^Cu. The specific activity of the ^64^Cu ranges from 47.4 to 474 GBq/μmol (1280 to 12,800 mCi/μmol) [66]. However, this method is limited by the high cost of ^64^Ni-enriched targets [67]. Additionally, several alternative routes for ^64^Cu production are reported, such as ^64^Zn(n,p) ^64^Cu [68] and ^63^Cu(n,γ) ^64^Cu [69]. Recently, Dellepiane et al. reported a ^64^Cu production via ^65^Cu(p,pn) ^64^Cu and ^67^Zn(p,α) ^64^Cu reactions [70].

In addition to theranostic properties, a longer half-life of cyclotron-produced copper-64 provides logistical advantages over clinically used cyclotron-produced fluorine-18 (T_1/2_ = 109 min) due to its ability to conduct PET studies away from the cyclotron; also, a longer half-life when compared to clinically used gallium-68 (T_1/2_ = 68 min) makes it possible to provide extended tumor imaging, as well as metastatic detection. The lower positron energy of ^64^Cu when compared to ^68^Ga provides a lower positron mean range (0.56 mm versus 3.5 mm), resulting in improved PET image quality, increased resolution, and higher diagnostic quality [71]. Additionally, a direct comparison of PET/CT image quality and spatial resolution obtained with ^18^F, ^68^Ga and ^64^Cu revealed a similar image quality obtained with ^18^F-FDG and ^64^Cu-HCl, evidently due to similar energies arising from the decay of the ^18^F and ^64^Cu positions. In contrast, the much higher energy of positrons arising from the decay of ^68^Ga degrades image quality and spatial resolution [72].

Copper-67 (T_1/2_ = 61.8 h, β^−^: 100%, γ: 49%) is the longest-lived radioisotope of copper, with a half-life that is suitable for imaging and beta particle energy that is appropriate for therapy. The tendency of isotopes to gamma decay with appropriate energies provides the possibility for simultaneous SPECT imaging and radiotherapy, as well as monitoring of the uptake and biodistribution of the radiotherapeutic agent during therapy [73]. A slightly higher β^−^ emission energy than that of clinically used ^177^Lu, in conjunction with a shorter half-life, causes ^67^Cu to be considered as an “ideal” radionuclide for radioimmunotherapy, which is commonly associated with slow pharmacokinetics [74]. However, despite the great potential for both imaging and therapy, worthwhile research of ^67^Cu potential as a theranostic radionuclide has not been carried out due to the inaccessibility of the radionuclide itself. 

Obtaining a sufficient amount of the ^67^Cu isotope with a high specific activity using accelerator-based production has always been a difficult task [75]. The commonly used proton-based reactions ^68^Zn(p,2p) ^67^Cu and ^70^Zn(p,α) ^67^Cu are complicated by the co-production of ^64^Cu. Recently, ^67^Cu production via the ^70^Zn(p,α) ^67^Cu reaction with a compact cyclotron was reported by Brühlmann et al. [76]. In addition, deuteron beam irradiation of enriched ^70^Zn target ^70^Zn(d,αn) ^67^Cu is a highly promising method for ^67^Cu production [77,78]. The neutron-induced reaction ^67^Zn(n,p) ^67^Cu requires a high-flux nuclear reactor with a fast-neutron flux [79]. A breakthrough of ^67^Cu production was achieved in the U.S. through the Department of Energy Isotope Program (DOE-IP). The use of high-energy photon-induced reactions on isotopically enriched ^68^Zn targets ^68^Zn(γ,p)^67^Cu opened opportunities for a reliable source of ^67^Cu with high specific activity > 1850 GBq/mg, radionuclide purity and sufficient quantity [80,81].

In addition to the above indisputable advantages of copper radionuclides, the well-established coordination chemistry of copper provides its reaction with different types of chelator systems [82]. The chemical identity of imaging ^64^Cu and therapeutic ^67^Cu copper radioisotopes allows for the convenient use of the same bifunctional chelators for both ^64^Cu and ^67^Cu-based radiopharmaceuticals, for sequential PET imaging and radiotherapy. Considering the opened opportunities for obtaining ^67^Cu of high purity and activity, an interest in the use of ^64^Cu/^67^Cu as a theranostic pair revived sharply (Figure 1).

Copper-based radiopharmaceuticals have been widely discussed for the treatment and diagnosis of various diseases in the last 30 years. 

[^64^Cu][Cu(ATSM)] is a group of thiosemicarbazone-based drugs, effective PET-tracers of tumor hypoxia, which proved its effectiveness in both preclinical and clinical studies. Still, one of the ^64^Cu ATSM drugs is in phase II of clinical trials for rectum cancer imaging [83]. In 2020, Liu et al. summarized the use of copper radiopharmaceuticals [Cu(ATSM)] for PET imaging of hypoxic tumors [84]. 

In 2020, we summarized the use of different metals, including ^64^Cu in PET imaging of Alzheimer’s disease [85]. In 2018, Ahmedova et al. summarized theranostic applications of copper radionuclides [86]. In 2023, Capriotti et al. summarized the use of ^64^CuCl_2_ for both PET imaging and theragnostic [87].

Herein, we summarize recent (2018–2023) advances in the use of copper-based radiopharmaceuticals, for PET imaging, radiotherapy and the use of ^64^Cu/^67^Cu as a theranostic pair.

## 2. Summary of Copper-Based Radiopharmaceuticals, Reported in 2018–2023

We summarized the copper-based radiopharmaceuticals developed in 2018–2023 in Table 2.

## 3. Copper-Based Radiopharmaceuticals Based on Peptides

### 3.1. Octreotate

Somatostatin (SST) is a small peptide that regulates both cell growth and hormone secretion. Somatostatin receptors (SSTRs) are a common target for the treatment of neuroendocrine tumors (NETs) [112]. Octreotate is a peptide capable of SSTRs binding, which is used worldwide for the targeted delivery of radioactive isotopes to NETs [113,114].

Detectnet (Copper ^64^Cu-dotatate) is a PET-imaging agent for the localization of SSTR-positive NETs in adult patients, which was approved by the Food and Drug Administration (FDA) in 2020 [88]. One year before, in 2019, a similar ^68^Ga-based radiopharmaceutical [^68^Ga]Ga-DOTA-TOC was approved by FDA as the first ^68^Ga-radiopharmaceutical for imaging of SSTR-positive gastroenteropancreatic NETs [115]. Both radiopharmaceuticals are based on tyrosine-octreotate, conjugated with DOTA (tetraxetan) as a metal chelator (Figure 2).

A direct comparison of Detectnet and [^68^Ga]Ga-DOTA-TOC, which was provided in 59 patients with NETs, showed undeniable advantages in lesion detection in NET patients of Detectnet over ^68^Ga-DOTATOC [116]. Recently, Song et al. reported a ^64^Cu-DOTATATE uptake in a 43-year-old woman with a slowly enlarging pulmonary nodule [89].

Cullinane et al. reported a preclinical investigation of a similar (Tyr3)-octreotate-based radiopharmaceutical, with MeCOSar as copper chelator, CuSarTATE (Figure 2) [90]. Two injections of [^67^Cu]CuSarTATE (15-MBq fractions two weeks apart) showed good antitumor efficacy in vivo on BALB/C mice with AR42J tumor, similar with [^177^Lu]LuTATE. Currently, [^67^Cu]CuSarTATE is ongoing clinical trials as drug for radionuclide therapy of neuroblastoma in pediatric patients (NCT04023331).

### 3.2. PSMA

Prostate-specific membrane antigen (PSMA) is a transmembrane glycoprotein that consists of 750 amino acids, that are overexpressed in tumor tissues 100- to 1000-fold higher than that in normal ones. The commonly used substrate of PSMA is a urea-based peptide with a C-terminal glutamate, capable of active site binding [117,118]. In 2022, Debnath et al. summarized PSMA-targeting and theranostic agents [119]. In 2022, Jeitner et al. also summarized advances in PSMA theranostics [120]. Herein, we provide several copper-based radiopharmaceuticals for prostate cancer therapy and imaging. 

Gallium (^68^Ga) gozetotide is a clinically used drug for PET prostate cancer imaging, which was approved in the United States in December 2020 [121] and in the European Union in December 2022 [122]. 

Despite FDA approval of (^68^Ga) gozetotide for prostate cancer imaging, several benefits of ^64^Cu over ^68^Ga arouse interest in ^64^Cu-based radiopharmaceuticals for PET imaging of prostate cancer [123]. Additionally, ^64^CuCl_2_ salt showed promising therapeutic efficacy on the 3D prostate cancer model [124].

In 2018, Umbricht et al. reported a successful PET visualization of PSMA-positive PC-3 PIP/flu tumor with ^64^Cu-based PSMA conjugates capable of albumin binding (Figure 3) [92].

PSMA-specific uptake for both radiolabeled ligands was confirmed on PSMA-positive (PC-3 PIP) and PSMA-negative (PC-3 flu) tumor cells. Biodistribution and PET/CT imaging studies performed on PC-3 PIP/flu tumor-bearing mice proved the ability of ^64^Cu−PSMA−ALB-89 to accumulate in PSMA-positive tumors. Even though ^64^Cu−PSMA−ALB-56 showed lower tumor uptake than ^64^Cu−PSMA−ALB-89, almost no retention of ^64^Cu−PSMA−ALB-56 was detected in the kidneys, which is critical to avoid false positive PET results. 

In 2020, Kelly et al. reported RPS-085 ligand, capable of both PSMA and serum albumin binding, and a theranostic approach based on ^64^Cu/^67^Cu pair for prostate cancer therapy and imaging (Figure 4) [92]. 

RPS-085 was designed based on previously reported ligand RPS-063 with DOTA chelator [125]. Metal-free RPS-085 showed the ability for both PSMA inhibition and human serum albumin binding. After ligand radiolabeling, LNCaP was successfully visualized with [^64^Cu]Cu-RPS-085. Importantly, the biodistribution of [^67^Cu]Cu-RPS-085 closely mimics that of [^64^Cu]Cu-RPS-085, thus confirming the possibility of pre-imaging when using the theranostic ^64^Cu/^67^Cu pair. Although studies of the therapeutic efficacy of [^67^Cu]Cu-RPS-085 have not been conducted, the principal possibility of pre-imaging with copper-64 before copper-67 radiotherapy has been proven. 

In 2019, Zia et al. reported two sarcophagine ligands with one or two PSMA-targeting moieties (Figure 5) [93]. The cell surface binding and internalization were evaluated in LNCaP cells and [^64^Cu]CuSarbisPSMA displayed higher cell surface binding and internalization, which is evidence that two target-binding vectors yield better results than one. 

To access imaging properties, PET images of PSMA-positive LNCaP-tumor-bearing NSG mice were obtained at 0.5, 2 and 22 h p.i., and significant tumor uptake of both radioligands was evident. Expectedly, bivalent [^64^Cu]CuSarbisPSMA showed higher tumor uptake and retention when compared to the monomer, which was confirmed in vivo biodistribution studies. However, [^64^Cu]CuSarPSMA showed better tumor uptake than clinically used ^68^Ga-PSMA (Ga-PSMA-11) at 1 h p.i. 

Since ^64^Cu-CuSarbisPSMA showed both an excellent uptake and retention in LNCaP tumors, the suggestion that the ^67^Cu variant may be suited to PSMA-targeted radiotherapy is relevant. Thus, in 2021, McInnes et al. reported a therapeutic potential of ^67^Cu-CuSarbisPSMA [94]. Expectedly, [^67^Cu]CuSarbisPSMA and [^177^Lu]LuPSMA provided similar tumor inhibition and survival extension at equivalent administered activities, since the energy from the β emissions from ^67^Cu and ^177^Lu are similar. However, the shorter half-life of ^67^Cu than of ^177^Lu (61.9 h vs. 6.7 days) shortens radiotherapy while maintaining its efficiency. ^64^Cu-SAR-bisPSMA and ^67^Cu-SAR-bisPSMA are currently in clinical trials as drugs for identification and treatment of PSMA-expressing metastatic castrate resistant prostate cancer (NCT04868604).

### 3.3. Other Peptides

In 2018, Sarkar et al. reported five bifunctional chelators, conjugated with arginyl glycyl aspartic acid (RGD peptide), and radiolabeled them with ^64^Cu. To evaluate the effects of the chelator’s nature on the pharmacokinetics of ^64^Cu-radiopharmateutical, five different chelators were used (Figure 6) [95].

Three ^64^Cu-labeled cross-bridged chelators showed better in vivo stability compared to the two non-cross-bridged chelators and ^64^Cu-labeled PCB-TE2A-Bn-NCS proved to be the most stable. PET imaging in glioma U87MG tumor-bearing mice was obtained, and two ^64^Cu-labeled PCB-TE2A conjugates exhibited higher tumor uptake compared with others. ^64^Cu-PCB-TE2A-Bn-NCS-c(RGDyK) also showed 4-fold lower demetallation in blood compared with the others. 

The melanocortin-1 receptors (MC1Rs) are a group of G protein-coupled receptors, which are overexpressed in human melanomas. MC1Rs can bind with alpha-melanocyte-stimulating hormone (α-MSH) peptides [126]. 

In 2022, Qiao et al. demonstrated the ability of copper-based radiopharmaceuticals to visualize melanoma. Two theranostic ^64^Cu-radiolabeled α-MSH peptides were designed as potential agents for melanoma PET imaging (Figure 7) [96].

Radiolabeled peptide ^64^Cu-NOTA-PEG2Nle-CycMSHhex showed both high MC1R binding affinity on B16/F10 cells and MC1R-specific cellular uptake on B16/F10 cells. Good tumor uptake of ^64^Cu-NOTA-PEG2Nle-CycMSHhex on B16/F10 melanoma-bearing mice was demonstrated by PET.

Gastrin-releasing peptide receptor (GRPR) is overexpressed on the surface of different cancers. GRPR can bind with high affinity to bombesin, a 14-amino acid peptide. Since bombesin itself exhibits low stability, its analogs have been investigated as GRPR-targeted ligands for the diagnosis and therapy of GRPR-positive tumors [127]. 

In 2022, Huynh et al. reported successful radiotherapy of GRPR-positive PC-3 tumor with ^67^Cu-labeled bombesin antagonist [97]. ^67^Cu-radiolabeled GRPR-targeted peptide [^67^Cu]Cu-SAR-BBN was designed (Figure 8).

[^67^Cu]Cu-SAR-BBN showed the ability to accumulate well in the GRPR-positive PC-3 cell line. Administration of six doses of 24 MBq of [^67^Cu]Cu-SAR-BBN resulted in inhibited PC-3 tumor growth with a 93.3% reduction in tumor volume, with no significant weight loss.

## 4. Copper-Based Radiopharmaceuticals for Radioimmunotherapy

### 4.1. Direct Conjugation of Radiolabeled Chelator and Antibody

Radioimmunotherapy (RIT) is a subtype of radiotherapy, that uses monoclonal antibodies as a delivery agent for radionuclides. Antibodies labeled with positron-emitting radionuclides are used for PET imaging and dosimetry, while radioimmunoconjugates labeled with therapeutic nuclides are used for therapy [128,129,130].

PD-1/PD-L1 inhibitors are a class of anticancer drugs, capable of blocking the activity of PD-1 and PDL1 immune checkpoint proteins [131]. Thus, anti-PD-1 or anti-PD-L1 antibodies are clinically used, and non-invasive imaging of PD-L1 expression levels in malignant tumors is of interest [132]. 

In 2018, Xu et al. reported the successful immunotherapy of a PD-L1 positive MC38 tumor with an anti-PD-L1 antibody, which was preliminary radiolabeled with copper-64, and its tumor accumulation was confirmed using PET [98] (Figure 9). 

When PET imaging of MC38 and 4T1 tumor grafts in vivo were performed, only the PD-L1 positive MC38 tumor was visualized by radiolabeled antibody [^64^Cu]Cu-NOTA-MX001. Immunotherapy studies provided in mice bearing MC38 tumor with MX001 antibody resulted in tumor growth suppression. In contrast, low anti-tumor efficacy of MX001 on 4T1 tumor was revealed, thereby proving the effectiveness and specificity of immunotherapy. Thus, an antibody may be successfully visualized with ^64^Cu before immunotherapy.

Trastuzumab is a human epidermal growth factor receptor protein (HER-2)-affine monoclonal antibody, clinically used in the treatment of (HER-2)+ metastatic breast cancer [133].

In 2021, Lee et al. reported a visualization of a NIH3T6.7 tumor with a ^64^Cu-radiolabeled trastuzumab antibody. In addition, a novel conjugation approach based on click reaction was proposed [99]. For the chemical binding of the antibody with a radiolabeled chelator, Tz/TCO click reaction was used. Tz/TCO is a bio-orthogonal inverse electron-demand Diels–Alder click reaction between trans-cyclooctene (TCO) and tetrazine (Tz) (Figure 10).

Copper-catalyzed azide–alkyne cycloaddition is usually not used in the synthesis of copper-chelating conjugates, due to it chelating the catalyst with reagents and the subsequent failure of the reaction. However, Lee et al. succeeded in choosing the conditions for the click reaction in which the catalytic agent is not chelated and the fast and quantitative click conjugation of the chelator and linker occurs. Since a cross-bridged chelator is not prone to complexation with Cu(II) ions at a lower temperature, Cu(I)-catalyzed alkyne−azide cycloaddition was used for conjugation of chelator and linker (Figure 11).

Both ^64^Cu-radiolabeled trastuzumab conjugates showed in vivo stability, and tumor accumulation and proved the ability to visualize a HER-2 positive NIH3T6.7 tumor. However, the conjugate with a PEG linker demonstrated fast body clearance.

Pertuzumab is another anti-HER-2 humanized monoclonal antibody that is used in combination with Trastuzumab in the therapy of HER-2-positive breast cancers [134].

In 2021, Hao et al. reported successful radioimmunotherapy of HER-2 positive HCC1954 tumor with radiolabeled ^67^Cu-Pertuzumab [100]. [^67^Cu]Cu-NOTA-Pertuzumab was obtained by conjugation of p-SCN-Bn-NOTA to pertuzumab and further radiolabeling. The efficacy of radioimmunotherapy was assessed on mice xenografts bearing a HER2 positive HCC1954 tumor. During the therapy, a dose-dependent tumor growth inhibition was observed even with the low dose of [^67^Cu]Cu-NOTA-Pertuzumab. 

A theranostic potential of ^67^Cu was confirmed via registration of SPECT/CT imaging after the injection of [^67^Cu]Cu-NOTA-Pertuzumab. Tumors were successfully visualized by SPECT, thereby confirming the possibility of the successful use of ^67^Cu radiopharmaceuticals as theranostic agents. 

CD4+ T cells are inflammatory mediators of autoimmune rheumatoid arthritis [135]. In 2022, Clausen et al. reported a visualization of rheumatoid arthritis with ^64^Cu-labeled radiotracer [101] (Figure 12).

The F(ab)’2 fragments of R-anti-mouse CD4 antibodies were conjugated to NOTA and radiolabeled with ^64^Cu. PET/CT images of a mouse with collagen-induced arthritis at different time points were obtained. Despite the drug accumulation in various organs, an increased accumulation of [^64^Cu]Cu-NOTA-IgG2b in joints with pronounced arthritis was revealed. Additionally, a decrease in tracer accumulation after dexamethasone injection confirmed a correlation of [^64^Cu]Cu-NOTA-CD4 accumulation with arthritic inflammation levels. 

### 4.2. Pretargeting Approach in Conjugation of Radiolabeled Chelator and Antibody

One of the main disadvantages of radioimmunotherapy is the fact that it can take several days for antibodies after administration to accumulate in their therapeutic target (tumor tissue). Thus, if antibodies are used as delivery agents for therapeutic radionuclides, only long-lived radionuclides should be used, which can lead to high radiation doses to healthy tissues. To solve this problem, in vivo pretargeting approach was suggested based on injecting the two components separately. An antibody is given several hours (or days) to accumulate in the tumor and clear from the blood. Then, the radiolabeled small molecule, capable of chemical binding with the antibody, is administered [136].

In 2020, Keinänen et al. reported an in vivo pretargeting with ^64^Cu/^67^Cu theranostic pair, with both PET imaging and subsequent radioimmunotherapy of SW1222 human colorectal carcinoma [102].

Two radioligands, [^64^Cu]CuMeCOSar-Tz and [^67^Cu]Cu-MeCOSar-Tz, as well as TCO-conjugated huA33 antibody, capable of targeting the A33 antigen, which is expressed in >95% colorectal cancers, were designed (Figure 13). 

Xenografts grafted with SW1222 human colorectal carcinoma (A33 antigen-positive) were administered with huA33-TCO, and after 24 or 72 h [^64^Cu]Cu-MeCOSar-Tz was injected. In the absence of huA33-TCO, [^64^Cu]Cu-MeCOSar-Tz showed negligible tumoral uptake, while in the mice treated with both huA33-TCO and [^64^Cu]CuMeCOSar-Tz, clear tumor PET imaging was registered. 

As for therapeutic efficacy, various strategies of longitudinal therapy have been tried to find the optimal dose and interval. As a result, HuA33-TCO and [^67^Cu]CuMeCOSar-Tz were injected 72 h apart, and a dose-dependent therapeutic response was observed. Importantly, PET images registered after injection of [^64^Cu]Cu-MeCOSar-Tz accurately predicted the efficacy of the [^67^Cu]Cu-MeCOSar-Tz, which was injected later, which is direct evidence of the effectiveness of the theranostic couple concept. 

In 2022, Jallinoja et al. reported another pretargeting approach, with novel ferrocene-based radioligands ([^64^Cu]Cu-NOTA-PEG3-Fc and [^64^Cu]Cu-NOTA-PEG7-Fc) (Figure 14) [103]. To conjugate antibodies with a chelator, a host–guest chemistry between a cucurbit [7] uril (CB7) and a ferrocene (Fc) was used [137]. 

M5A, CB7-M5A antibody can bind carcinoembryonic antigen (CEA), which is expressed in several cancers, such as colorectal, gastric and pancreatic cancers, and also in some breast cancer and non-small-cell lung cancer [138]. The antibody was modified with dibenzocyclooctyne. Both radioligands showed good in vitro stability and had similar in vivo profiles in healthy mice, with relatively slow excretion through the gastrointestinal tract. The pretargeting approach has been investigated with a time interval of 120 h, and radioligands showed specific tumor uptake. In addition, a pretargeting approach with an extended time interval of up to 9 days still showed good tumor localization.

## 5. Another Copper-Based Radiopharmaceutical

Boron neutron capture therapy (BNCT) is based on the irradiation of boron-10-based agents with low-energy thermal neutrons to yield high yields of lithium-7 and alpha particles. The heavy alpha particle has a short range, which allows for the localization of the radiation effect [139]. However, despite the advantages of BNCT, mapping boron-based biodistribution in the patient is unobtainable [140]. For boron mapping, optical imaging and PET imaging may be applied. One of the ways for both imaging methods to be optimized is the use of boronated porphyrins, which can chelate copper cations, resulting in 64Cu-based agents for visualization [141].

In 2018, Shi et al. reported a successful visualization and subsequent BNCT of a 4T1 tumor with 64Cu-radiolabeled micelle-coated boronated porphyrins (Figure 15) [104]. 

Fluorescence imaging properties of BPN were confirmed in vivo in 4T1 tumor-bearing mice, and tumor imaging was performed with the system. Tumors were visualized separately from surrounding tissues.

After labeling BPN with copper-64, both high accumulation and long retention in the tumor were confirmed in PET images of B16–F10 tumor-bearing mice. Finally, complete tumor suppression in mice-bearing B16–F10 tumors administrated with BPN after neutron irradiation confirmed the effectiveness of BNCT. Preliminary PET visualization of boronated porphyrin with ^64^Cu radiopharmaceuticals is a novel perspective approach, allowing control of the distribution of the drug and the localization before treatment.

Earley et al. reported two photoactive pro-ligands H_2_ATSM/en-ArN_3_ and H_2_ATSM-PEG_3_-ArN_3_, capable of light-induced photochemical bioconjugation produce with ^64^Cu-radiolabelled protein (Figure 16) [105]. Radiolabeled [^64^Cu]Cu-H_2_ATSM/en-ArN_3_ and [^64^Cu]Cu-H_2_ATSM-PEG_3_-ArN_3_ were obtained via either direct synthesis or transmetallation of corresponding Zn(II) complexes. The light-induced reaction of the aryl azide group yielded the ^64^Cu-radiolabelled HSA proteins with azepine linker.

These data open up a novel possibility for easy and quick radiolabeling of biomolecules with ^64^Cu/^67^Cu via using a photochemistry approach, thus yielding radiopharmaceuticals for PET imaging for radiotherapy.

### Nanoparticles (Nps)

Nps are a powerful tool for various biomedical applications such as targeted drug delivery, bioimaging, diagnostics, theranostics and therapy for various diseases [142,143,144,145]. Among the countless applications of nanoparticles, their use for combined MRI-PET diagnostics, achieved by labeling Nps with imaging or theranostic isotopes is of interest [146]. Thus, an introduction of copper-64 radioactive isotope makes it possible to obtain materials for both PET diagnostics and radiotherapy of tumors. Currently, there is a clinical trial phase 1 under recruiting to evaluate ^64^Cu-labeled NPs to guide the surgical treatment of prostate cancer (NCT04167969).

Recently, Pijeira et al. summarized the use of radiolabeled nanomaterials for biomedical applications [147]. Recently, Carrese et al. have also summarized the use of Nps in cancer theranostics [148]. Herein, we provide several examples of radiolabeled Nps for various biomedical applications. 

In 2018, Madru et al. reported a hybrid PET-MRI probe, based on ^64^Cu-radiolabeled superparamagnetic iron oxide Nps. In addition to a simple radiolabeling technique that does not require a chelating agent, successful double PET-MRI imaging of the lymph nodes has been performed [106].

Chelator-free radiolabeled [^64^Cu]CuS Nps have also been discussed as a promising agent for simultaneous micro-PET/CT imaging and photothermal ablation therapy [149]. In 2018, Cai et al. reported [^64^Cu]CuS Nps, coated with bombesin peptide to enhance tumor accumulation via specific uptake (Figure 17) [107].

Expectedly, Bom-PEG-[^64^Cu]CuS NPs showed both specific binding to prostate cancer cells and enhanced cellular uptake. Additionally, Bom-PEG-[^64^Cu]CuS NPs successfully visualized prostate cancer and demonstrated enhanced tumor uptake when compared to PEG-[^64^Cu]CuS, which was confirmed by both micro-PET/CT and biodistribution studies. 

Thakare et al. reported a trifunctional imaging probe, based on AGuIX^®^ Nps [150], functionalized with NODAGA copper chelator, NIR heptamethine cyanine dye and maleimide as stabilized moiety (Figure 18) [108].

The resulting AGuIX^®^ nanoparticles, functionalized with IR-783-Lys(Mal)NODAGA are appropriate for simultaneous PET-MRI and optical trimodal imaging, which was confirmed in a TSA tumor model.

In 2020, Zhou et al. reported a ^64^Cu-labeled PEGylated melanin Nps, which was previously reported as a promising platform for multimodality imaging [109,151]. Radiolabeled melanin Nps showed a therapeutic effect on the A431 tumor. 

In 2020, Paiva et al. also reported a polymeric micellar Nps (PMNPs), conjugated with EGFR-targeting GE11 peptide via diazo-tyrosin coupling [110]. Importantly, the prelabeling strategy was used in radiolabeling of Nps with ^64^Cu (Figure 19).

^64^Cu-GE11 PMNPs displayed enhanced tumor accumulation due to EGFR targeting effects, which was confirmed by both PET imaging and biodistribution studies in vivo EGFR-positive colorectal HCT116 tumor model.

He et al. have reported antigen-delivery nanoplatforms based on poly(ethylene glycol) (PEG, Mw 500) and pyropheophorbide-A (PPa) in order to deliver the melanoma antigen peptide, Trp2180−188 (SVYDFFVWL), to dendritic cells (DCs) and stimulate CD8+ T-cell immune response [111] (Figure 20).

DC uptake of Trp2/PPa−PEGm was confirmed using flow cytometry. Radiolabeling of NPs with ^64^Cu allows real-time monitoring of the migration process of labeled DCs to draining lymph nodes (DLNs), which was demonstrated in C57BL/6 mice.

In addition, a vaccine based on DCs treated with Trp2/PPa−PEGm NPs stimulated a significant immune response in C57BL/6 mice. Finally, a significant tumor growth inhibition was detected in C57BL/6 mice with B16-F10 melanoma tumor, injected with DCs/Trp2/PPa−PEGm NPs three times at a weekly interval. This result not only demonstrates the possibility of immunotherapy with NP-modified DCs but also the possibility of biodistribution monitoring in vivo after radiolabeling with ^64^Cu. With the use of the ^67^Cu isotope, a combination of immunotherapy and radiotherapy would be possible, which is of interest.

## 6. Conclusions

In 2020, copper ^64^Cu-dotatate was approved by the FDA as a radioactive diagnostic agent for PET-imaging agent for SSTR-positive NETs in adult patients. The mere fact of the approval of a copper-containing drug for clinical practice encourages researchers to design new effective copper-containing drugs for the therapy, diagnosis and theragnostic of various diseases. Due to the unique emission properties of copper isotopes, there is great interest in their use as both imaging and therapeutic agents. Copper-64 is a cyclotron-produced nuclide with excellent energy characteristics and optimum half-life, allowing for thorough PET imaging of malignant neoplasms not available with the short-lived ^68^Ga and ^18^F nuclides.

The ^67^Cu isotope has long been regarded as an “ideal but inaccessible” radionuclide for radiotherapy and radioimmunotherapy, due to its excellent energy characteristics and long half-life. A recent breakthrough in the production of copper-67 isotopes made it possible to essay this previously inaccessible radionuclide in action, both as a nuclide for radiotherapy and for theranostics. Additionally, the long half-life of the copper-67 isotope makes it an ideal nuclide for radioimmunotherapy, and for control of the accumulation of antibodies in a therapeutic target.

The use of the theranostic pair copper 64/67 is also of great interest due to the convenient interchangeability of copper ions. Since the nuclide ^67^Cu isotope was previously not available in sufficient quantities, a ^64^Cu/^67^Cu was not tested as a theranostic pair either. Now, the opportunity for sequential PET imaging, dosimetry, radiotherapy and SPECT imaging has opened up.

In this review, we have summarized a recent successful application of copper-based radiopharmaceuticals for PET, SPECT imaging, radiotherapy, and radioimmunotherapy. Thus, several successful PET-imaging of malignant neoplasms with ^64^Cu-based radiopharmaceuticals were reported [91,92,95,96], as well as imaging of rheumatoid arthritis [81], and visualization of the distribution of agents for boron neutron capture therapy [83]. A principal possibility of pre-imaging with copper-64 before radiotherapy with copper-67 has been proven by Kelly et al. [92]. 

In addition, several successful radiotherapies with therapeutic radionuclide ^67^Cu-based agents were reported. Thus, McInnes et al. reported the therapeutic efficacy of ^67^Cu-CuSarbisPSMA in prostate cancer therapy [94], ^67^Cu-radiolabeled bombesin antagonist peptide [^67^Cu]Cu-SAR-BBN was effective in PC-3 tumor therapy, as reported by Huynh et al. [97]. 

Several quite interesting results in PET imaging with antibody-based agents, PET visualization of antibody biodistribution, immunotherapy, and radioimmunotherapy were also summarized. A successful PET pre-imaging of antibody accumulation in tumor, followed by effective immunotherapy, was reported by Xu et al. [98], PET-visualization of HER-2 positive NIH3T6.7 tumor with ^64^Cu-labeled trastuzumab antibody was reported by Lee et al. [99]. An extremely important result, namely, a successful radioimmunotherapy of HER-2 positive HCC1954 tumor with SPECT imaging, was reported by Hao et al. [100]. These data are evidence of both the radiotherapeutic properties of the copper-67 isotope and the possibility of the use of ^67^Cu-based radiopharmaceuticals as theranostic agents.

A pretargeting approach based on separate injections of the antibody and radiolabeled chelator has shown its effectiveness both for imaging of and therapy for tumor diseases [102,103]. An important and elegant study was reported by Keinänen et al. [102]. demonstrated the use of ^64^Cu/^67^Cu theranostic couple in the pretargeting assay; not only successful PET imaging of the SW1222 human colorectal carcinoma with ^64^Cu-based conjugate but also therapeutic efficacy of ^67^Cu-based conjugate has been shown. PET images predicted the efficacy of radiotherapy, which is direct evidence of the effectiveness of the theranostic couple concept. 

The development of radiolabeled nanoparticles is definitely of interest, due to the possibility of simultaneous use of several diagnostic modalities, such as PET-MRI, fluorescence imaging, SPECT-MRI, etc. Additionally, the radiolabeling of nanoparticles with copper-67 beta-emitter is of great interest for the development of theranostic platforms. 

Several incredibly successful results in the therapy, diagnosis and theranostic of tumor diseases, presented in this review, show the great potential of copper-containing radiopharmaceuticals in nuclear medicine and medicinal chemistry. Given the recent breakthrough in obtaining the copper-67 isotope in sufficient quantity and purity, the field of use of both the therapeutic radionuclide ^67^Cu and the theranostic pair ^64^Cu/^67^Cu is just beginning; however, the results obtained so far are quite impressive. Thus, both the effectiveness and great potential of copper-containing radiopharmaceuticals both as imaging and therapy agents are undoubted.

## Figures and Tables

**Figure 1 ijms-24-09154-f001:**
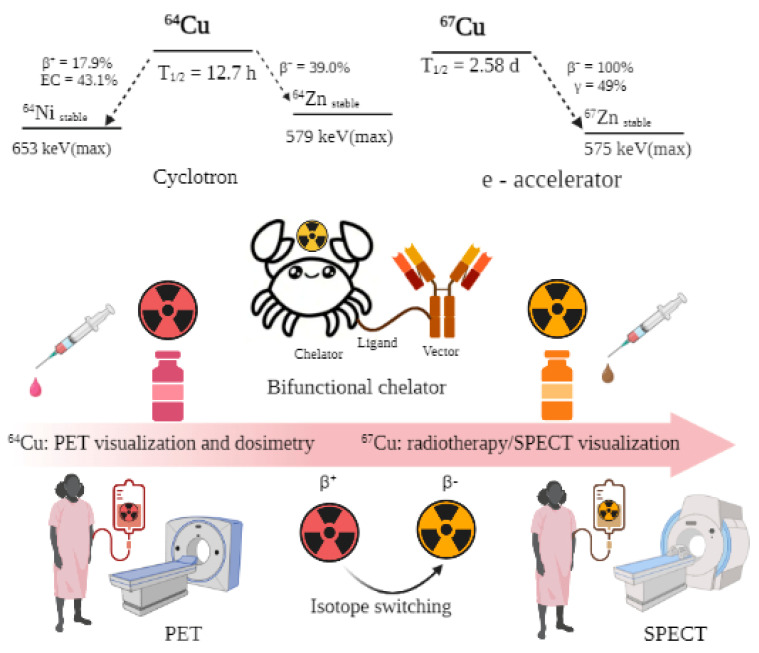
Schemes of decay of ^64^Cu and ^67^Cu radionuclides, and a principal scheme of using ^64^Cu/^67^Cu theranostic pair for preimaging, dosimetry and consequence radiotherapy.

**Figure 2 ijms-24-09154-f002:**
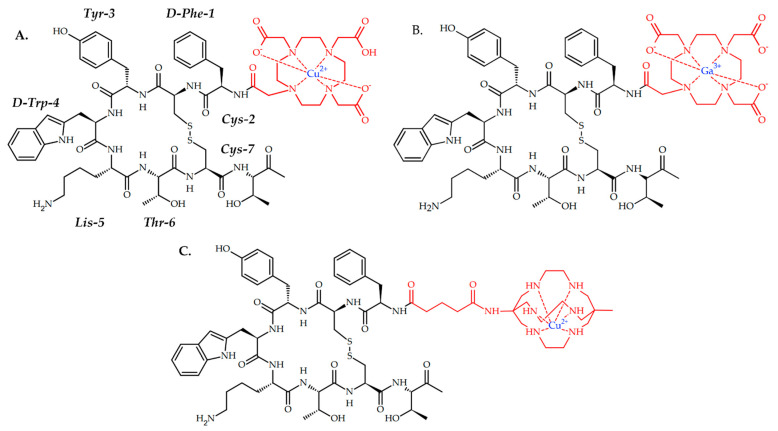
(**A**) Detectnet, ^64^Cu-based radiopharmaceutical for PET imaging of SSTR positive NETs; (**B**) [^68^Ga]Ga-DOTA-TOC, ^68^Ga-based radiopharmaceutical for PET imaging of SSTR positive NETs in adult and pediatric patients. (**C**) CuSarTATE, ^67^Cu-based radiopharmaceutical for radionuclide therapy of neuroblastoma in pediatric patients.

**Figure 3 ijms-24-09154-f003:**
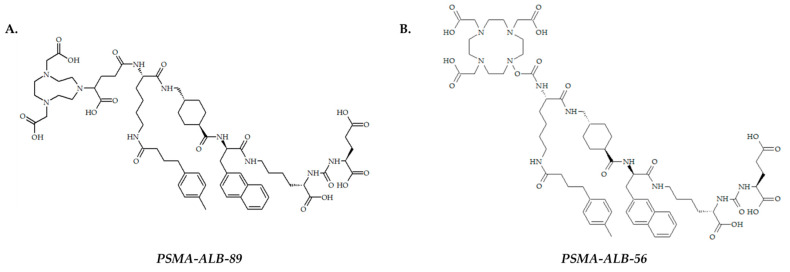
PSMA-ALB-89 (**A**) and PSMA-ALB-56 (**B**), reported by Umbricht et al. [91].

**Figure 4 ijms-24-09154-f004:**
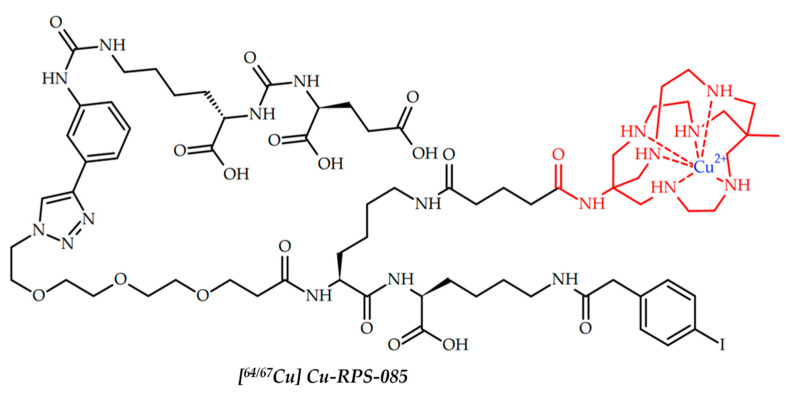
[^64^Cu]Cu-RPS-085, reported by Kelly et al. [92].

**Figure 5 ijms-24-09154-f005:**
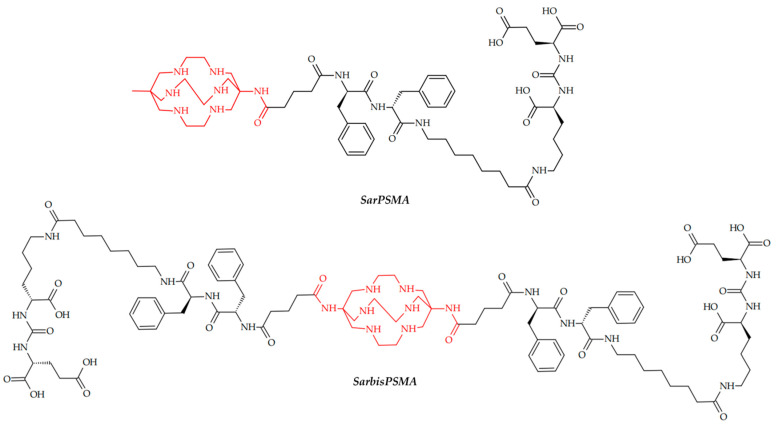
SarPSMA and SarbisPSMA, reported by Zia et al. [93].

**Figure 6 ijms-24-09154-f006:**
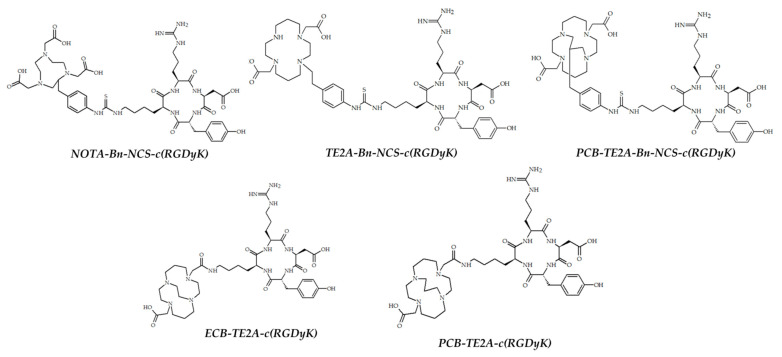
BFCs, reported by Sarkar et al. [95].

**Figure 7 ijms-24-09154-f007:**
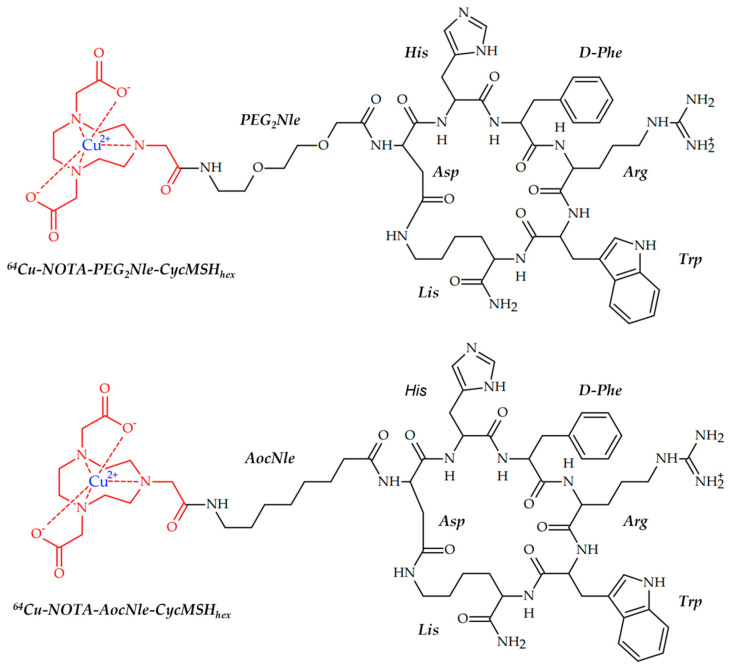
MC1Rs-affine radiolabeled peptides for melanoma imaging, reported by Qiao et al. [96] A. ^64^Cu-NOTA-PEG2Nle-CycMSHhex; B. ^64^Cu-NOTA-AocNle-CycMSHhex.

**Figure 8 ijms-24-09154-f008:**
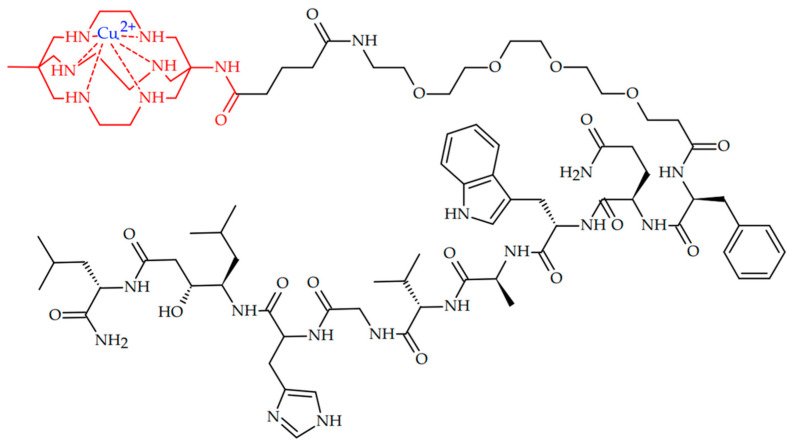
^67^Cu-radiolabeled bombesin antagonist peptide [^67^Cu]Cu-SAR-BBN, reported by Huynh et al. [97].

**Figure 9 ijms-24-09154-f009:**
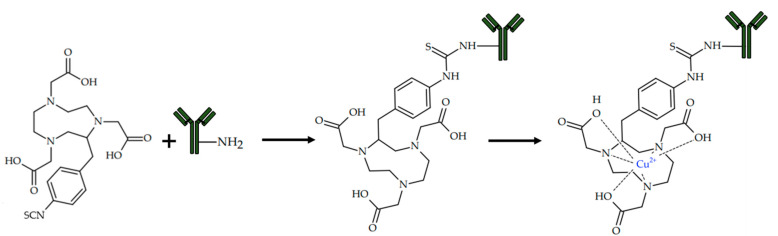
Chemical structure of [^64^Cu]Cu-NOTA-MX001, reported by Xu et al. [98].

**Figure 10 ijms-24-09154-f010:**
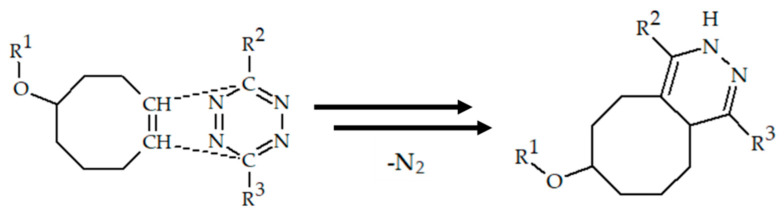
TCO—tetrazine conjugation, a commonly used approach for quick chemical binding of antibody.

**Figure 11 ijms-24-09154-f011:**
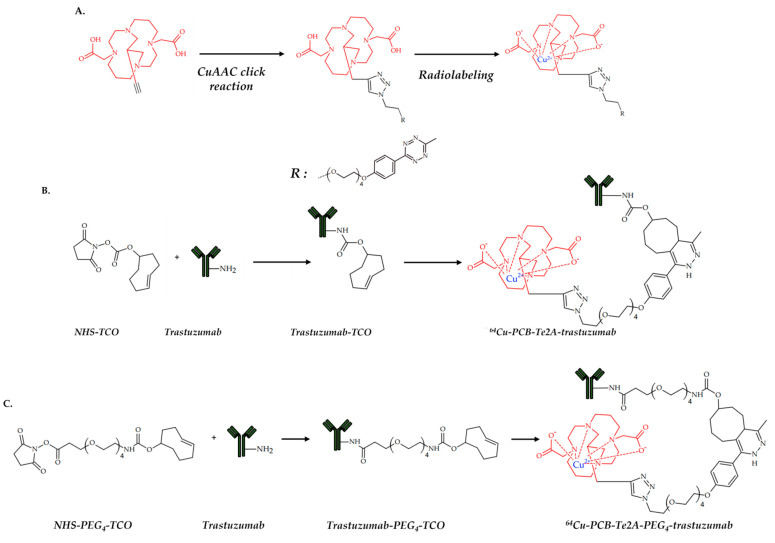
Cross-bridged chelators with radioactive copper ions (**A**), and its conjugation with trastuzumab (**B**,**C**), reported by Lee et al. [99].

**Figure 12 ijms-24-09154-f012:**
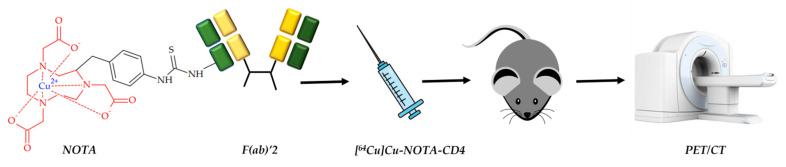
^64^Cu-labeled radiotracer based on F(ab)’2 fragments of R-anti-mouse CD4 antibodies and NOTA as copper chelator, reported by Clausen et al. [101].

**Figure 13 ijms-24-09154-f013:**
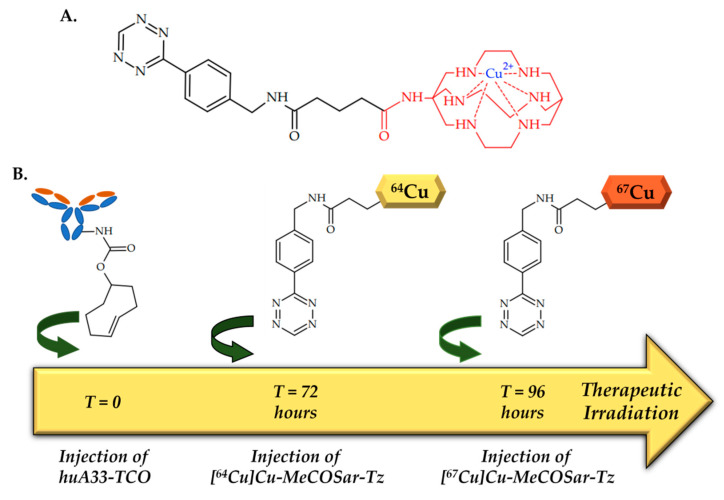
(**A**) Chemical structure of Cu-MeCOSar-Tz. (**B**) Design of the experiment, performed by Keinänen et al. [102].

**Figure 14 ijms-24-09154-f014:**
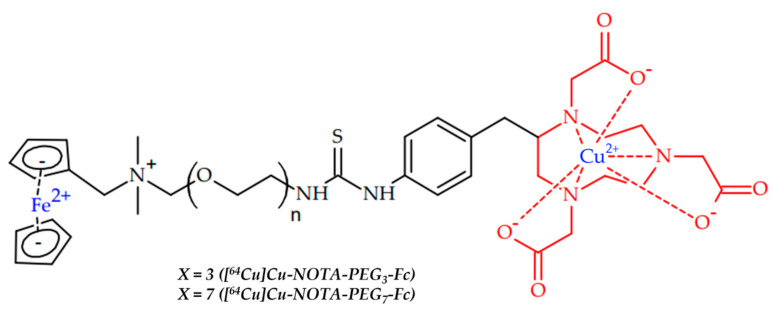
Ferrocene-based radioligands ([^64^Cu]Cu-NOTA-PEG_3_-Fc and [^64^Cu]Cu-NOTA-PEG_7_-Fc), reported by Jallinoja et al. [103].

**Figure 15 ijms-24-09154-f015:**
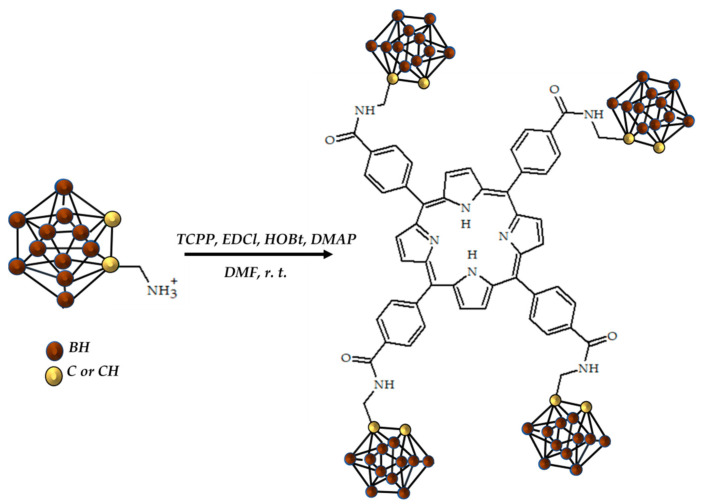
Boronated porphyrin, designed by Shi et al. [104].

**Figure 16 ijms-24-09154-f016:**
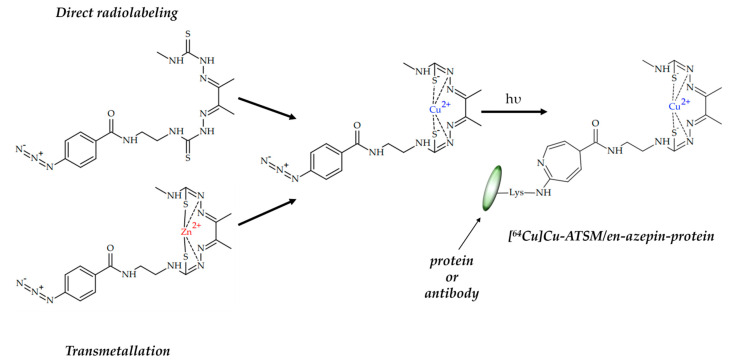
Photoactive pro-ligands H_2_ATSM/en-ArN_3_ and H_2_ATSM-PEG_3_-ArN_3_, and their light-induced photochemical bioconjugation with protein, designed by Earley et al. [105].

**Figure 17 ijms-24-09154-f017:**
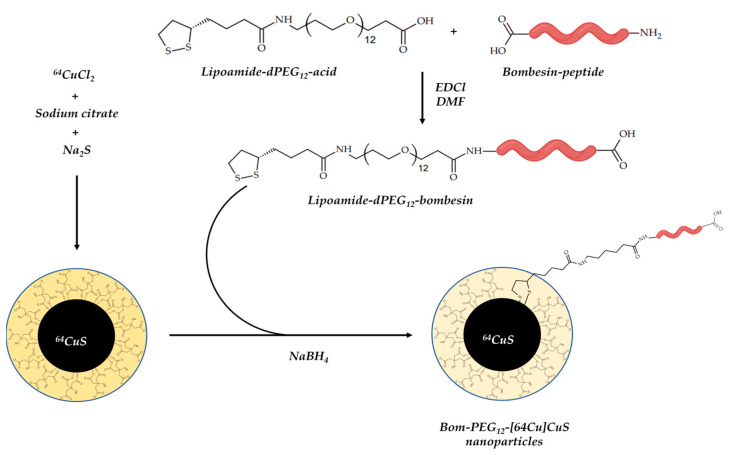
Bom-PEG-[^64^Cu]CuS Nps, reported by Cai et al. [107].

**Figure 18 ijms-24-09154-f018:**
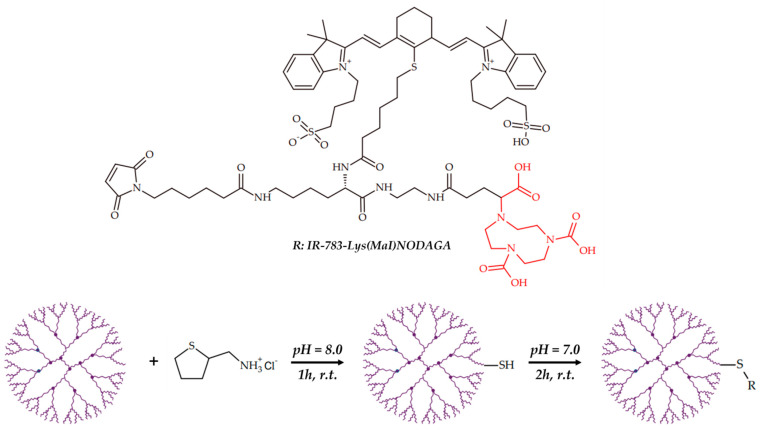
AGuIX^®^ nanoparticles, functionalized with IR-783-Lys(Mal)NODAGA, reported by Thakare et al. [108].

**Figure 19 ijms-24-09154-f019:**
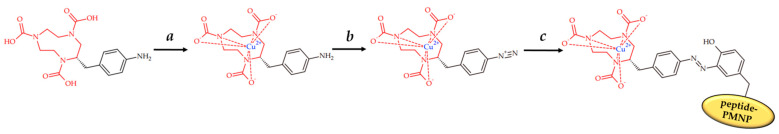
^64^Cu-labeled PMNPs, designed by Paiva et al. [110] Reagents and conditions: (**a**) [^64^Cu]CuCl_2_, 0.1 M NH_4_OAc (pH 5.5), 15 min, 37 °C; (**b**) NaNO_2_, HCl, pH 1, 5 min, 4 °C; (**c**) GE11 or HW12 PMNPs, 0.1 M borate buffered saline (pH 8−9), 15 min, 4 °C.

**Figure 20 ijms-24-09154-f020:**
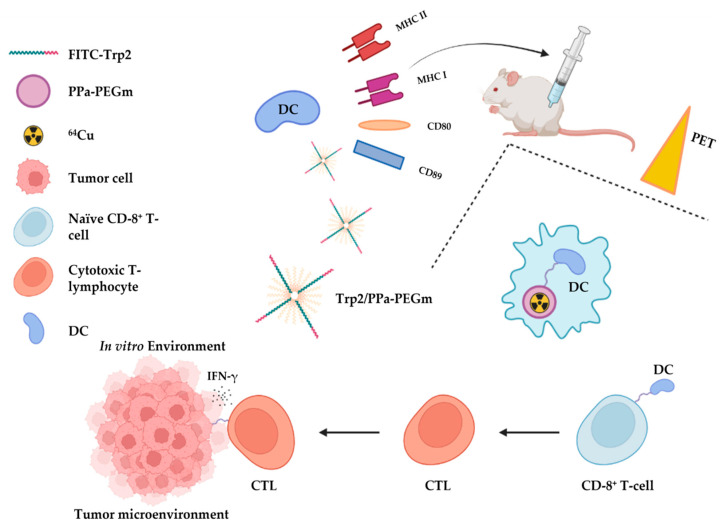
PEG-PPa Trp2 melanoma antigen-delivery nanoplatform, radiolabeled with ^64^Cu, designed by He et al. [111].

**Table 1 ijms-24-09154-t001:** Radionuclides for the therapy and diagnosis of various diseases, and their clinical applications.

Positron-Emitting Radionuclides						
Nuclide	Decay	Energy, keV (Intensity)	Production	Drugs	Target/Application	Ref.
Copper-62 (^62^Cu)	T_1/2_ = 9.76 min β^+^ = 97.8% EC = 2.2%	β^+^ mean energy 1319: 2937 (max, 97.6%), 1321 (av); γ: 511 (195.6%);	^62^Zn/^62^Cu generator Cyclotron ^61^Ni(p,γ)^62^Cu	[^62^Cu]ATSM [^62^Cu]PTSM	Tumor hypoxia	[8]
Carbon-11 (^11^C)	T_1/2_ = 20.4 min β^+^ = 99.8%	β^+^ mean energy 385.7: 960 (max, 99.8%), 385.7 (av); γ: 511 (195.5%);	Cyclotron ^14^N(p,α)^11^C	[^11^C]Flumazenil	GABA, epilepsy imaging	[9]
				[^11^C]Raclopride	D2 dopamine receptors, Parkinson Disease imaging	[10]
				[^11^C]Methionine	Neuroncology, tumor imaging	[11]
				[^11^C]Choline	Prostate cancer	[12]
				[^11^C]Pittsburgh Compound B	Alzheimer disease	[13]
Copper-60 (^60^Cu)	T_1/2_ = 23.7 min β^+^ = 93% EC = 7%	β^+^ mean energy 970: 1912 (max, 11.6%), 839.6 (av); 1982 (max, 49.0%), 872.0 (av); 2947 (max, 15.0%), 1104 (av); γ: 511 (185.0%), 826.4 (21.7%), 1332.5 (88.0%), 1791.6 (45.4%);	Cyclotron ^60^Ni(p,n)^60^Cu	[^60^Cu]ATSM	Tumor hypoxia	[14]
Fluorine-18 (^18^F)	T_1/2_ = 109.8 min β^+^ = 96.7% EC = 3.3%	β^+^ mean energy 249.8: 633.5 (max, 96.7%), 249.8 (av); γ: 511 (193.5%);	Cyclotron ^18^O(p,n) ^18^F	[^18^F] FDG (^18^F-2-fluoro-2-deoxy-D-glucose)	Tumors Neurological disorders Inflammation	[15]
				[^18^F]FLT (^18^F-3′-fluoro-3′-deoxythymidine	Thymidine kinase-1 (TK-1) imaging biomarker of cellular proliferation Cancer	[16]
Gallium-68 (^68^Ga)	T_1/2_ = 67.7 min β^+^ = 88.9% EC = 11.1%	β^+^ mean energy 829.5: 1899 (max, 87.7%), 836.0 (av); γ: 511 (177.8%), 1077 (3.22%)	^68^Zn(p,n) ^68^Ga ^68^Ge/^68^Ga generator	[^68^Ga]PSMA-11; [^68^Ga]DOTA-TATE; [^68^Ga]DOTA-TOC; [^68^Ga]DOTA-NOC; [^68^Ga]FAPI-04	Prostate cancer imaging SSTR(+) neuroendocrine tumors imaging Fibroblast-activation-protein—positive tumor imaging	[17,18,19,20]
Copper-61 (^61^Cu)	T_1/2_ = 3.34 h β^+^ = 61% EC = 39%	β^+^ mean energy 500: 932.8 (max, 5.8%), 399.0 (av); 1216 (max, 51.0%), 524.0 (av); γ: 511 (123%), 283.0 (12.7%), 656.0 (10.4%), 1185.2 (3.6%);	Cyclotron ^61^Ni(p,n)^61^Cu ^60^Ni(d,n)^61^Cu	[^61^Cu]ATSM [^61^Cu]APTS	Tumor hypoxia	[21]
Scandium-43	T_1/2_ = 3.89 h β^+^ = 88% EC = 23%	β^+^ mean energy 476: 825.8 (max, 17.2%), 344.5 (av); 1199 (max, 70.9%), 508.1 (av); γ: 511 (176.2%), 372.9 (22.5%);	^43^Ca(p,n)^43^Sc ^44^Ca(p,2n)^43^Sc ^42^Ca(d,n)^43^Sc	-	-	[22]
Scandium-44	T_1/2_ = 4.04 h β^+^ = 94.2% EC = 5.8%	β^+^ mean energy 632: 1473 (max, 94.3%), 632 (av); γ: 511 (188.5%), 1157 (99.9%);	Cyclotron ^44^Ca(d,2n)^44^Sc Generator ^44^Ti/^44^Sc ^nat^Ca(p,n)^44^Sc	-	-	[23,24]
Copper-64 (^64^Cu)	T_1/2_ = 12.7 h β^+^ = 17.5% EC = 43.5% β^−^ = 39.0%	β^+^ mean energy 278: 652.6 (max, 17.5%), 278.0 (av); β^−^ mean energy 190.7: 579.6 (max, 38.5%), 190.7 (av); γ: 511 (35.0%), 1346 (0.47%);	Cyclotron ^64^Ni(p,n)^64^Cu ^64^Ni(d,2n)^64^Cu Reactor ^64^Zn(n,p)^64^Cu ^63^Cu(n,γ)^64^Cu	Copper (^64^Cu) oxodotreotide	Somatostatin receptor positive neuroendocrine tumors	[25]
Zirconium-89 (^89^Zr)	T_1/2_ = 3.27 d EC = 77% β^+^ = 23%	β^+^ mean energy 396: 902 (max, 22.7%), 395.5 (av); γ: 511 (45.5%), 909.2 (99.0%);	Cyclotron ^89^Y(p,n) ^89^Zr ^89^Y(d,2n)^89^Zr	^89^Zr-labeled a mAbs	Targeting depends on mAbs type	[26,27,28,29,30]
Iodine-124 (^124^I)	T_1/2_ = 4.2 d β^+^ = 23% EC = 77%	β^+^ mean energy 820: 1535 (max, 11.7%), 687.0 (av); 2138 (max, 10.7%), 974.7 (av); γ: 511 (45%), 602.7 (62.9%), 722.8 (10.4%), 1691 (11.2%);	Cyclotron ^124^Te(d,2n)^124^I ^124^Te(p,n)^124^I	Thyroid cancer Radiolabeling mAbs	Differentiated thyroid cancer (DTC)	[31]
**Gamma-emitting radioisotopes**						
Technetium-99m (^99^mTc)	T_1/2_ = 6.0 h IT = 100%	γ: 140.5 (89%)	^99^Mo/^99m^Tc generators	Tc-^99^m medronate	Bone imaging agent	[32]
				Tc-^99^m arcitumomab	Carcinoembryonic antigen (CEA) Colorectal tumors imaging	[33]
					Hepatocytes, hepatobiliary imaging	[34]
Iodine-123 (^123^I)	T_1/2_ = 13.2 h EC = 100%	γ: 159.0 (83.6%), 529.0 (1.27%);	Cyclotron ^124^Xe(p,pn) ^123^Xe ^123^I Accelerator ^123^Te(p,n)^123^I	Sodium iodide-123	Thyroid cancer imaging	[35]
				Ioflupane (^123^I)	Dopamine transporter binding, diagnosis of Parkinson’s disease	[36]
				Lofetamine (^123^I)	Non-specific receptor binding. Cerebral blood perfusion imaging	[37]
				Lomazenil (^123^I)	Benzodiazepine antagonist GABA receptors imaging	[38]
				Lobenguane (^123^I)	Noradrenaline transporters Pheochromocytomas Neuroblastomas	[39]
Thallium-201 (^201^Tl)	T_1/2_ = 3.04 d EC = 100%	γ: 68.9 (26.6%), 70.8 (44.7%), 80.2 (10.3%), 167.4 (10.0%);	Cyclotron	Thallous chloride	Myocardial imaging	[39]
Gallium-67 (^67^Ga)	T_1/2_ = 3.26 d EC = 100%	γ: 93.3 (38.8%), 184.6 (21.4%), 300.2 (16.6%), 393.5 (4.6%);	Cyclotron ^68^Zn(p,2n)^67^Ga	Gallium citrate	Primary and metastatic tumors	[40]
Indium-111 (^111^In)	T_1/2_ = 2.8 d EC = 100%	γ: 171.3 (90.7%), 245.4 (94.1%);	Cyclotron ^112^Cd(p,2n) ^111^In ^111^Cd(p,n) ^111^In	Indium (^111^In) capromab pendetide	Prostate cancer	[41]
				Antibody labeling	Lymphoma	[42]
**β^−^-emitting radioisotopes**						
Samarium-153 (^153^Sm)	T_1/2_ = 1.9 d β^−^ = 100%	β^−^ mean energy 225: 634.6 (max, 30.4%), 199.5 (av); 704.3 (max, 49.2%), 225.2 (av); 264.3 (max, 19.5%), 807.5 (av); γ: 69.7 (4.7%), 103.2 (29.1%);	Reactor ^152^Sm (n, γ) ^153^Sm	^153^Sm-ethylene diamine tetramethylene phosphonate (EDTMP)	Bone pain palliation	[43]
Copper-67 (^67^Cu)	T_1/2_ = 2.58 d β^−^ = 100%	β^−^ mean energy 141: 377.1 (max, 57%), 121 (av); 468.4 (max, 22.0%), 154 (av); 189 (max, 20.0%), 561.7 (av); γ: 91.3 (7.0%), 93.3 (16.1%), 184.6 (48.7%);	Cyclotron ^70^Zn(p,α)^67^Cu ^67^Zn(n,p)^67^Cu Accelerator ^68^Zn(p,2p)^67^Cu ^70^Zn(d,αn)^67^Cu and ^68^Zn(γ,p)^67^Cu	-	Radioimmunotheranostics	[44]
Yttrium-90 (^90^Y)	T_1/2_ = 2.6 d β^−^ = 100%	β^−^ mean energy 932.3: 2279 (max, 100%), 932.4 (av);	^90^Sr/^90^Y generator	TheraSphere	Radioembolization of hepatocellular carcinoma	[45]
Scandium-47 (^47^Sc)	T_1/2_ = 3.35 d β^−^ = 100%	β^−^ mean energy 162: 440.9 (max, 68.4%), 142.6 (av); 600.3 (max, 31.6%), 203.9 (av); γ: 159.4 (68.3);	^46^Ca(n,γ)^47^Ca → ^47^Sc ^47^Ti(n,p)^47^Sc	-	-	[46]
Rhenium-186 (^186^Re)	T_1/2_ = 3.7 d β^−^ = 92.5% EC = 7.5%	β^−^ mean energy 348: 936 (max, 21.48%), 307.4 (av); 1073 (max, 70.9%), 360.5 (av); γ: 137.15 (9.47%);	Reactor ^185^Re(n,γ) ^186^Re	Rhenium-186 HEDP	Palliative treatment of bone metastases	[47]
				Re-186-labeled sulfur colloid	Therapy of rheumatoid arthritis	[48]
Xenon-133 (^133^Xe)	T_1/2_ = 5.2 d β^−^ = 100%	β^−^ mean energy 100.3: 266.8 (max, 1.4%), 75.16 (av); 346.4 (max, 98.5%), 100.6 (av); γ: 81.9 (36.9%);	By-product resulting from nuclear reactors	Gas	Lung perfusion tests	[49]
Lutetium-177 (^177^Lu)	T_1/2_ = 6.7 d β^−^ = 100%	β^−^ mean energy 133.6: 175.5 (max, 11.7%), 47.2 (av); 383.9 (max, 8.9%), 111.2 (av); 496.8 (max, 79.4%), 148.8 (av); γ: 112.9 (6.23%), 208.4 (10.4%);	^176^Lu (n, γ)^177^Lu ^176^Yb(n,γ)^177^Yb→^177^Lu	[^177^Lu]Lu-DOTA-TATE	SSTR-expressing tumors	[50]
Iodine-131 (^131^I)	T_1/2_ = 8.0 d β^−^ = 100%	β^−^ mean energy 181.9: 247.9 (max, 2.08%), 69.4 (av); 333.8 (max, 7.23%), 96.6 (av); 606.3 (max, 89.6%), 191.6 (av); γ: 284.3 (6.12%), 364.5 (81.5%), 637 (7.16%), 722.9 (1.77%);	By-product resulting from nuclear reactors ^235^ U(n,f)^131^I	Sodium iodide	Treat hyperthyroidism (an overactive thyroid)	[51]
Strontium-89 (^89^Sr)	T_1/2_ = 51 d β^−^ = 100%	β^−^ mean energy 587.1: 1501 (max, 100%), 587.1 (av);	Thermal reactor ^88^Sr (n,γ) ^89^Sr ^89^Y (n,p)^89^Sr	Strontium-89 chloride	Metastatic bone lesions imaging	[52]
**α-emitting radioisotopes**						
Bismuth-213 (^213^Bi)	T_1/2_ = 45.6 min α = 2.14% β^−^ = 97.86%; (to ^213^Po, then ^209^ Pb via α decay)	β^−^ mean energy 436: 982 (max, 30.1%), 320 (av); 1422 (max, 66.8%), 491.8 (av); γ: 440.5 (25.9%); α: 5875 (1.96%), 8376 (98%, ^213^Po);	Part of the ^225^Ac decay	-	-	[53]
Lead-212 (^212^Pb)	T_1/2_ = 10.6 h β^−^ = 100%; decays to α-emitters ^212^Bi and ^212^Po	β^−^ mean energy 101.3: 153.8 (max, 5.0%), 40.88 (av); 330.5 (max, 81.5%), 93.28 (av); 569.1 (max, 13.7%), 171.4 (av); γ: 238.6 (43.6%); α: 6051 (25.1%, ^212^Bi), 6090 (9.8%, ^212^Bi), 8785 (64.0%, ^212^Po);	^224^Ra/^212^Pb generator	^212^Pb-labelled DOTAMTATE	SSR-binding, therapy of SSR-positive tumors	[54]
Actinium-225 (^225^Ac)	T_1/2_ = 9.9 d α = 100% decays to ^213^Bi	α: 5637 (4.4%), 5724 (3.1%), 5732 (8.0%), 5792 (18.1%), 5830 (50.7%);	Th-229/Ac-225 generators ^226^Ra(n,γ)^227^Ra ^226^Ra(p,2n)^225^Ac	^225^Ac-PSMA-617	Prostate cancer therapy	[55,56]
				Actinium-225 Nitrate	Antibody labeling	[57]
Thorium-227 (^227^Th)	T_1/2_ = 18.7 d	α: 5708 (8.3%), 5757 (20.4%), 5979 (23.5%), 6038 (24.2%);	Part of the ^227^Ac decay series	-	Antibody labeling	[58]

**Table 2 ijms-24-09154-t002:** Copper-based radiopharmaceuticals, reported in 2018–2023.

**Compound**	**Chelator**	**Targeting Moiety/Target**	**Ref.**
**Copper-based radiopharmaceuticals based on peptides**			
Octreotate			
[^64^Cu]Cu-DOTA-TOC [^64^Cu]Cu-DOTA-TATE [^67^Cu]Cu-SARTATE	DOTA MeCOSar	Octreotate Somatostatin receptors (Neuroendocrine tumors)	[88,89,90]
[^64^Cu]Cu-PSMA-ALB-89 [^64^Cu]Cu-PSMA-ALB-56	NODAGA DOTA	PSMA	[91]
[^64^Cu]Cu-RSP-085 [^67^Cu]Cu-RSP-085	MeCOSar	PSMA	[92]
[^64^Cu]CuSarbisPSMA	MeCoSAr BisCoSar	PSMA	[93]
[^67^Cu]CuSarbisPSMA	BisCoSar	PSMA	[94]
[^64^Cu]Cu-NOTA-BnNCS-c(RGDyK) [^64^Cu]Cu-TE2A-BnNCS-c(RGDyK) [^64^Cu]Cu-PCB-BnNCS-c(RGDyK) [^64^Cu]Cu-ECB-BnNCS-c(RGDyK) [^64^Cu]Cu-PCB-TE2A-c(RGDyK)	NOTA TE2A PBC-TE2A EBC-TE2A	RGD peptide/Integrine	[95]
[^64^Cu]Cu-NOTA-PEG2Nle-CycMSHhex [^64^Cu]Cu-NOTA-AocNle-CycMSHhex	NOTA	α-MSH peptide/ Melanocortin-1 receptor	[96]
[^67^Cu]Cu-SAR-BBN	SAR	Bombesin antagonist peptide/ Gastrin-releasing peptide receptor	[97]
**Copper-based radiopharmaceuticals for radioimmunotherapy**			
Direct conjugation of radiolabeled chelator and antibody			
[^64^Cu]Cu-NOTA-MX001	NOTA	Anti-PD-L1 antibody/PD-L1	[98]
[^67^Cu]Cu-PCB-TE2A-PEG-4-Trastuzumab	PCB-TE2A	Trastuzumab/HER2	[99]
[^67^Cu]CuNOTAPertuzumab	NOTA	Pertuzumab/HER2	[100]
[^64^Cu]CuNOTACD4	NOTA	R-anti-mouse CD4 antibodies/CD4	[101]
Pretargeting approach in conjugation of radiolabeled chelator and antibody			
[^64^Cu]Cu-MeCOSar-Tz [^67^Cu]Cu-MeCOSar-Tz	MeCOSar	HuA33 antibody/A33 antigen	[102]
[^64^Cu]Cu-NOTA-PEG3-Fc [^64^Cu]Cu-NOTA-PEG7-Fc	NOTA	CB7-M5A antibody/Carcinoembryonic antigen	[103]
**Another copper-based radiopharmaceuticals**			
[^64^Cu]Cu-boronated porphyrins	Porphyrin	-	[104]
[^64^Cu]Cu-H2ATSM/en-ArN3 [^64^Cu]Cu-H2ATSM-PEG3-ArN3	ATSM	-	[105]
Nanoparticles (Nps)			
^64^Cu-radiolabeled superparamagnetic iron oxide Nps	Chelator-free	-	[106]
[^64^Cu] CuS Nps	Chelator-free	Bombesin peptide	[107]
AGuIX^®^ Nps	NODAGA	Maleimide	[108]
Melanin Nps		-	[109]
Polymeric micellar Nps	NOTA	EGFR-targeting GE11 peptide	[110]
PEG-PPa Nps	Pyropheophorbide-A	Dendritic cells (DCs)	[111]

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
