# Peer review of "Recent Advances in 64Cu/67Cu-Based Radiopharmaceuticals"

_ijms, 2023, doi:10.3390/ijms24119154_

Round 1

Reviewer 1 Report

This paper is an in-depth review on 64Cu and 67Cu radiopharmaceuticals for theranostics application. The review is well-structured. The authors consistently reviewed the main aspects of 64Cu and 67Cu radiopharmaceuticals. There are several errors that do not affect the overall assessment of the work. However, there are a number of questions and comments presented below:

1) The general problem of the article is that the issues of production of 64Cu and 67Cu isotopes and their comparison with others are not presented. Despite all the advantages of 64Cu and 67Cu, working with solid-state cyclotron targets during the production of these isotopes severely limits their scope. For example, the 18F isotope for PET will always have an advantage over 64Cu due to the ease of producing 18F in a liquid cyclotron target. In addition, it is necessary to compare the nuclear physical characteristics of 64Cu in comparison with 18F and 68Ga mentioned in the article.

2) In my opinion, the authors should also mention other medical isotopes of copper. In particular, 61Cu is considered more promising because it can be obtained by irradiating natural zinc, rather than the expensive 64Ni.

3) Table 1 has numerous of typos, shortcomings and factual errors. Following are just a few:

11C: Correct names for [11C]Flumazenil and [11C]Raclopride

68Ga: not a single radiotracer is shown, which is rather strange for one of the most valuable isotopes for PET. One could mention [68Ga]PSMA-11; [68Ga]DOTA-TATE; [68Ga]DOTA-TOC; [68Ga]DOTA-NOC; [68Ga]FAPI-04 and many other radiopharmaceuticals.

123I: Most often, 123I is produced by the reaction 123Te(p;n)123I

177Lu: Production of 177Lu (especially for TRT) proceeds by the reaction 176Yb(n,γ)177Yb→177Lu

225Ac: Exposure to 226Ra does not occur in a nuclear reactor for 225Ac. Usually for production of 227Ac or 228Ac.

4) Line 59. The authors made a gross mistake. Earlier, in Table 1, it was mentioned that 68Ga is obtained from a generator. With a generator in the clinic, you have access to this isotope every 3-4 hours. The problem with generators is only limited activity from generators. It is this fact, and not the short half-life, that is the limitation for 68Ga.

5) Line 85. The proposal is rather controversial due to the lack of data on 67Cu therapy. From the side of nuclear-physical characteristics, a short half-life is not always a plus. With a shorter half-life, more activity per injection is often required. In addition, reference 63 clearly says that there is a big problem with the availability of 67Cu. The availability of ~1.7-2 Ci per irradiation is too low for a wide coverage of therapy, especially due to the high availability and low cost of 177Lu.

6) Line 60. It is not clear how Reference 98 about the first clinical use of 64Cu-DOTA-TATE relates to the discussion about the advantage of 64Cu-PSMA.

Author Response

  • The general problem of the article is that the issues of production of 64Cu and 67Cu isotopes and their comparison with others are not presented. Despite all the advantages of 64Cu and 67Cu, working with solid-state cyclotron targets during the production of these isotopes severely limits their scope. For example, the 18F isotope for PET will always have an advantage over 64Cu due to the ease of producing 18F in a liquid cyclotron target. In addition, it is necessary to compare the nuclear physical characteristics of 64Cu in comparison with 18F and 68Ga mentioned in the article.

We thank you for your careful reading of the manuscript, and for your valuable comments. We have added a comparison of the copper-64 isotope with fluorine-18 and gallium-68.

  • In my opinion, the authors should also mention other medical isotopes of copper. In particular, 61Cu is considered more promising because it can be obtained by irradiating natural zinc, rather than the expensive 64

Thank you for this valuable comment. We have added information on other copper isotopes both to the main text of the manuscript and to Table 1.

3) Table 1 has numerous of typos, shortcomings and factual errors. Following are just a few:

  • 11C: Correct names for [11C]Flumazenil and [11C]Raclopride

Corrected.

  • 68Ga: not a single radiotracer is shown, which is rather strange for one of the most valuable isotopes for PET. One could mention [68Ga]PSMA-11; [68Ga]DOTA-TATE; [68Ga]DOTA-TOC; [68Ga]DOTA-NOC; [68Ga]FAPI-04 and many other radiopharmaceuticals.

Thank you for this valuable comment. We have added this necessary information and references.

  • 123I: Most often, 123I is produced by the reaction123Te(p;n)123I

Thank you for this valuable comment. We have added this necessary information to Table 1.

  • 177Lu: Production of 177Lu (especially for TRT) proceeds by the reaction 176Yb(n,γ)177Yb→177Lu

Thank you for this valuable comment. We have added this necessary information to Table 1.

  • 225Ac: Exposure to 226Ra does not occur in a nuclear reactor for 225Ac. Usually for production of 227Ac or 228Ac.

Thank you for this valuable comment. We have added this necessary information to Table 1.

  • Line 59. The authors made a gross mistake. Earlier, in Table 1, it was mentioned that 68Ga is obtained from a generator. With a generator in the clinic, you have access to this isotope every 3-4 hours. The problem with generators is only limited activity from generators. It is this fact, and not the short half-life, that is the limitation for 68

Thank you for your careful reading this valuable comment. We have added this necessary information to the main text of the manuscript; we have also deleted the information about short half-life issue of 68Ga. 

  • Line 85. The proposal is rather controversial due to the lack of data on 67Cu therapy. From the side of nuclear-physical characteristics, a short half-life is not always a plus. With a shorter half-life, more activity per injection is often required. In addition, reference 63 clearly says that there is a big problem with the availability of 67 The availability of ~1.7-2 Ci per irradiation is too low for a wide coverage of therapy, especially due to the high availability and low cost of 177Lu.

Thank you for your valuable comment. We fully agree that a short isotope half-life is not always a plus, and have changed our careless statement. We have also added absolutely necessary information about existing and new ways to obtain the isotope of copper-67. Currently, the copper-67 isotope is available in several ways for medical applications [Journal of Radioanalytical and Nuclear Chemistry volume 321, pages125–132 (2019)].

  • Line 60. It is not clear how Reference 98 about the first clinical use of 64Cu-DOTA-TATE relates to the discussion about the advantage of 64Cu-PSMA.

Thank you for your valuable comment. We have changed Ref. 98.

Reviewer 2 Report

The manuscript needs a complete revision. It's full of typos, mispellings and come concepts are poorly written. It is recommended a full restyling.

Clinical trials such as DISCO, CL04 and SECuRE should be discussed as these are the trials that are currently spearheading the future of clinical Cu64/Cu67 theranostics. 

Considerations on the productions methods and supply chains of Cu-60, -61, -62, -64, 67 should be laid out.

Gozetotide was approved in US in 2020 and not in 2021. Illuccix (ref 69) is not the first approved. The approval came from UCLA/UCSF in December 2020.

Very difficult to read and needs to be extensively edited and proofread. It is not publishable in this form.

Author Response

The manuscript needs a complete revision. It's full of typos, mispellings and come concepts are poorly written. It is recommended a full restyling.

Thank you for your careful reading. We have improved the language and style of the manuscript.

Clinical trials such as DISCO, CL04 and SECuRE should be discussed as these are the trials that are currently spearheading the future of clinical Cu64/Cu67 theranostics. 

We sincerely apologize, but we did not quite understand what clinical trials are proposed for discussion. We used to search Clinicaltrials.gov source. The only drug that we managed to find is the use of Disulfiram for the treatment of Covid-19, but it does not apply to radiopharmaceuticals. Please kindly add more information about which drugs we should discuss, and we will gladly do so.

Considerations on the productions methods and supply chains of Cu-60, -61, -62, -64, 67 should be laid out.

Thanks for the valuable comment. We have added information on obtaining the above copper isotopes to Table 1.

Gozetotide was approved in US in 2020 and not in 2021. Illuccix (ref 69) is not the first approved. The approval came from UCLA/UCSF in December 2020.

Thanks for the valuable comment. We have corrected this information.

Reviewer 3 Report

Table 1

PET radionuclides

Scandium isotopes does not belong to the most widely used radiometals. Together with Zr-89 they are rather in a "research and development stage".

Section β- emitting radioisotopes

For the Copper-67, I am missing here and in the introduction part, the 70Zn(d,αn)67Cu production reaction that provides one of the highest purity 67Cu isotope product, see the works of Kozempel et al. and Nigron et al.: https://doi.org/10.1524/ract.2012.1939, https://doi.org/10.1524/ract.2011.1879, https://doi.org/10.3389/fmed.2021.674617).

Figure 1

The term isotope exchange should not be used in this meaning of isotope switching, since it refers to specific chemical isotope exchange reactions - e.g. 65Cu for 67Cu. I would suggest to use other formulation. E.g. using theranostic pair of isotopes, like 64/67Cu. 

Summary of copper-based radiopharmaceuticals

Here I am missing the simple CuCl2 compound application, see e.g.: 

https://doi.org/10.3389/fmolb.2020.609172, https://doi.org/10.3390/jcm12010223

I am also missing the synthetic nanoparticulate and polymeric vectors labelled with Copper isotopes, e.g.: https://doi.org/10.2967/jnumed.107.045302,  https://doi.org/10.1524/ract.2009.1669,

https://doi.org/10.1186/s41181-022-00161-4, 

https://doi.org/10.1021/bc900511j, 

etc.

Please check and correct the format of the references. 

Author Response

  • Table 1

PET radionuclides

Scandium isotopes does not belong to the most widely used radiometals. Together with Zr-89 they are rather in a "research and development stage".

Thanks for this comment. We fully agree that scandium isotopes are not widely used, however, we considered it necessary to mention them due to the large number of articles on the development of radiopharmaceuticals based on scandium isotopes.

To avoid misunderstanding, we have changed the name of the Table 1.

  • Section β- emitting radioisotopes

For the Copper-67, I am missing here and in the introduction part, the 70Zn(d,αn)67Cu production reaction that provides one of the highest purity 67Cu isotope product, see the works of Kozempel et al. and Nigron et al.: https://doi.org/10.1524/ract.2012.1939, https://doi.org/10.1524/ract.2011.1879, https://doi.org/10.3389/fmed.2021.674617).

Thank you for your careful reading. We have added this information and references to the manuscript.

  • Figure 1

The term isotope exchange should not be used in this meaning of isotope switching, since it refers to specific chemical isotope exchange reactions - e.g. 65Cu for 67Cu. I would suggest to use other formulation. E.g. using theranostic pair of isotopes, like 64/67Cu. 

  • Summary of copper-based radiopharmaceuticals

Here I am missing the simple CuCl2 compound application, see e.g.: 

https://doi.org/10.3389/fmolb.2020.609172, https://doi.org/10.3390/jcm12010223

Thank you for this valuable comment. We have added this references to the manuscript.

I am also missing the synthetic nanoparticulate and polymeric vectors labelled with Copper isotopes, e.g.: https://doi.org/10.2967/jnumed.107.045302 

https://doi.org/10.1524/ract.2009.1669

https://doi.org/10.1186/s41181-022-00161-4, 

Thank you for your valuable comment. We have added a section on the use of radiolabeled nanoparticles for diagnostics and therapy.

Please check and correct the format of the references. 

Checked.

Reviewer 4 Report

This review article discusses the advances of copper radioisotopes, copper-64 and copper-67 related to their radiopharmaceutical applications from 2018-2023. It is a thorough review of the literature.

The authors could take into consideration the following comments:

1.  Information on the specific activities on copper-64 and copper-67 nuclides would be quite useful if provided in the introduction section.

2. In Figure 2, the structure of DOTA-TOC should be rechecked, particularly regarding the terminal threoninol. Also, the coordination number of Copper and gallium complexes probably wouldn’t be  greater than 6.

3. The text on HBED-CC ligand and the related structure of 68Ga-gozetotide should not be included since it is not the subject of this review.

4. Regarding the ligands in figure 4, they are also albumin binding ones, although it is not mentioned in the text. The structure of the PSMA-targeted ligands could be explained in more depth, regarding the pharmacophore unit and the albumin-binding unit.

5. Paragraph 3.3 Another peptides should be corrected to “Other peptides”

6. Lines 273-277 that describe Figure 11 should be better explained. What is linker R that is conjugated via click? Also why is the TCO-tetrazine conjugation method not described at this point, but is described later in the pretargeting method?

7. There are quite a few references from websites, eg drugbank, EMA etc. This should be avoided to such extent. The relevant information could be found from articles in the literature. It is recommended to revise.

1.The text would require English editing, and there are a few typing errors for revision.

2.The phrase: “Despite FDA approval of (68Ga) gozetotide for prostate cancer imaging, several dis-158 cussed above factors that pointed the benefits of 64Cu over 68Ga arouse interest in 64Cu-159 based radiopharmaceuticals for PET imaging of prostate cancer” was not well understood.

3. The phrase: “ In 2018, Sarkar et al. have reported five bifunctional chelators, conjugated arginyl-208 glycylaspartic acid (RGD peptide), and radiolabeled them with 64Cu to evaluate the effects 209 of chelator’s nature on the pharmacokinetic of 64Cu-radiopharmateutical” was not well understood.

Author Response

This review article discusses the advances of copper radioisotopes, copper-64 and copper-67 related to their radiopharmaceutical applications from 2018-2023. It is a thorough review of the literature.

The authors could take into consideration the following comments:

  1. Information on the specific activities on copper-64 and copper-67 nuclides would be quite useful if provided in the introduction section.

Thank you for this valuable comment. We have added this information.

  1. In Figure 2, the structure of DOTA-TOC should be rechecked, particularly regarding the terminal threoninol. Also, the coordination number of Copper and gallium complexes probably wouldn’t be greater than 6.

Thank you for the comment. The structure rechecked, as well as another structures to avoid any mistakes.

  1. The text on HBED-CC ligand and the related structure of 68Ga-gozetotide should not be included since it is not the subject of this review.

Thank you for this valuable comment. We have deleted this section.

  1. Regarding the ligands in figure 4, they are also albumin binding ones, although it is not mentioned in the text. The structure of the PSMA-targeted ligands could be explained in more depth, regarding the pharmacophore unit and the albumin-binding unit.

Thank you for this valuable comment. We have this information to the manuscript.

  1. Paragraph 3.3 Another peptides should be corrected to “Other peptides”

Corrected.

  1. Lines 273-277 that describe Figure 11 should be better explained. What is linker R that is conjugated via click? Also why is the TCO-tetrazine conjugation method not described at this point, but is described later in the pretargeting method?

Thank you for your valuable comment. We moved the mention of TCR-tetrazine conjugation.

  1. There are quite a few references from websites, eg drugbank, EMA etc. This should be avoided to such extent. The relevant information could be found from articles in the literature. It is recommended to revise.

Thank you for this valuable comment. We have revised this references.

Comments on the Quality of English Language

1.The text would require English editing, and there are a few typing errors for revision.

2.The phrase: “Despite FDA approval of (68Ga) gozetotide for prostate cancer imaging, several dis-158 cussed above factors that pointed the benefits of 64Cu over 68Ga arouse interest in 64Cu-159 based radiopharmaceuticals for PET imaging of prostate cancer” was not well understood.

Thank you for this valuable comment/ We have corrected this phrase.

  1. The phrase: “In 2018, Sarkar et al. have reported five bifunctional chelators, conjugated arginyl-208 glycylaspartic acid (RGD peptide), and radiolabeled them with 64Cu to evaluate the effects 209 of chelator’s nature on the pharmacokinetic of 64Cu-radiopharmateutical” was not well understood.

Thank you for this valuable comment. We have corrected this phrase.

Round 2

Reviewer 2 Report

SECuRE: NCT 04868604 PSMA

CL04: NCT 04023331 Sartate

Please search for these and add.

Author Response

Thank you for reading the manuscript carefully, and for pointing out important clinical trials. We have added these clinical trials to pages 9 and 12.